# Preserving the Unique Heritage of Chinese Ancient Architecture in Diffusion Models with Text and Image Integration

## Abstract

Leveraging the impressive generative capabilities of diffusion models, we can create diverse images from imaginative prompts with careful design. To be noticed, the key components, such as CLIP, are essential for aligning prompts with image representations. However, these models often underperform in specialized areas, like the Chinese ancient architecture. One of the important reasons is that historical buildings include not only architectural information, but also historical and cultural content. The preservation and integration of these unique characteristics has become a significant challenge in model expansion. In this paper, we propose an Image-Annotation-Augmented Diffusion pipeline combining human feedback to explore the specific-area paradigm for image generation in the context of small amounts of data and professional concepts. We first leverage Segment Anything 2 (SAM2) to obtain a refined content image to enable an in-depth analysis of the relationship between unique characteristics and multimodal image generation models, and reselected representative images and regrouped them according to their distinctive objective and the existing dataset. Then, we introduce the effective RAG and GraphRAG module to identify the complex structure of relationships among different entities in the training and inference stages respectively. Based on the initial text by BLIP3, the RAG instructs GPT4 to facilitate more accurate, content-aware annotations during training, and augment a high-quality object prompt using the GraphRAG during inference. Benefit from these outstanding models and architectures, we train fine-tuning models to showcase the enhanced performance of our proposed pipeline compared to other existing models. Experiments demonstrate that our pipeline effectively preserves and integrates the unique characteristics of ancient Chinese architecture.

## 1 Introduction

The development of generative models, like OpenAI (2023); Team et al. (2023); Li et al. (2022); Podell et al. (2023), has triggered revolutionary changes in the field of artificial intelligence. These models, built on Transformer Vaswani (2017) and Diffusion Ho et al. (2020) architectures and trained on diverse and extensive datasets, have demonstrated unprecedented capabilities in understanding, interpreting, and generating human language Peng et al. (2024) and ideal images Li et al. (2024). Especially for image-generating task, various satisfied results can be obtained by different language prompts. Outstanding performance of text-to-image models demonstrate unprecedented creative capabilities with realistic quality and a variety of images based on some prompt written in natural language Ramesh et al. (2022); Saharia et al. (2022); Ruiz et al. (2023). Hence, a lot of novel applications, including image Avrahami et al. (2022); Chen et al. (2024a), music Fei et al. (2024), text-to-speech Huang et al. (2022), are being developed based on the outstanding abilities of AI models.

The remarkable language comprehension and image-generating capabilities come from several aspects. In detail, the basic one is the massive corpora and gallery with a huge amount of high-quality data covering most universal contents. As mentioned in Zhuang et al. (2024), the large language models (LLMs) learn huge amounts of knowledge from enormous and diverse corpora. In the image generating area, as said in Dai et al. (2023), the outstanding performance comes from the profes-

sional image dataset. To effectively apply a satisfied tuning strategy, thousands of high-quality images and associated text are enough to cause a significant impact on the aesthetics of the generated images. Besides, developments of multimodel deep learning models, like CLIP Radford et al. (2021) and T5 Raffel et al. (2020), in text-to-image contributes to the boosting improvements. Several researches develop creative architectures and theories to promote numerous methodological and application innovations that significantly expand the scope and boost the functionality of diffusion models. Despite existing generative models perform satisfied in well-studied scene, it still face a noticeable issue that how to fine-tune a diffusion model to task- specific scenarios, like Chinese ancient architecture.

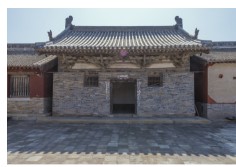 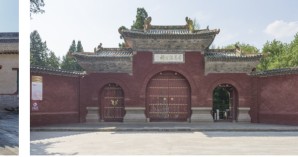 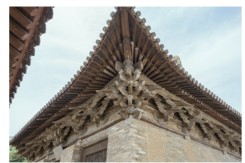 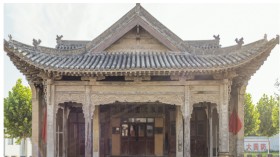

The image is the side hall which is rebuilt in the Yuan Dynasty, three bays wide and four rafters deep, with a single-eaved roof. The temple's architecture highlights evolving styles and represents an important relic of Chinese heritage. Zhengjue Temple, in Kausi Village, is a national-protected Buddhist temple rebuilt in the Jin, Yuan, and Ming Dynasties. Its gray-brick buildings feature intricate carvings, green-tiled eaves, and classical Chinese design.

This image shows the entrance to the Sima Wengong Temple, dedicated to Sima Guang, a renowned historian and politician from Xia County. The structure features red brick walls, intricate carvings, and a traditional Chinese gray-tiled roof. The gateway has three doors, with the central one larger than the side ones. Sima Guang, author of Zizhi Tongjian, was posthumously titled Grand Tutor Wen Guogong after his death in 1086. The building showcases classic Chinese architectural elements with red walls and gray roof tiles.

This image depicts a section of the Sanjiao Hall, Dongfeng Wanshou Palace, located in Shangdongfeng Village, Gaoping City. The building features intricate wooden carvings and ornate, upward-curving tiled eaves, typical of traditional Chinese temples. Founded in 1284 and later repaired during the Jin, Ming, and Qing dynasties, the temple showcases the rich architectural heritage of ancient China, with inscriptions inside that document its history.

This image shows the front view of the Sanguan Temple Stage, an ancient structure in Sanluli Village, Yanhu District, Yuncheng City, Shanxi Province. Built before the Yuan Dynasty, as indicated by a Yuan-era stele, it was later repaired in 1520, 1637, 1666, and 1842. The stage is associated with folk Taoism and is recognized as part of the fourth batch of provincial protected cultural sites. It reflects the architectural heritage of the Yuan and Ming Dynasties.

Figure 1: There are four example of Chinese ancient architectures. Both the images and their annotations are provided for better understanding the unique feathers and culture backgrounds.

In a specified area, training and fine-tuning strategies face bidirectional problems between data and models, arising from annotations, special entities, hierarchic content, cross-modal alignment, etc. The main reason refers to the fact that the task-specific properties require both domain knowledge and AI expertise Shen et al. (2023). Taking the Chinese ancient architecture as an example, as introduced in Li et al. (2023), there are a lot of unique linguistic features and cultural background information that result in great challenges for fine-tuning tasks. Especially when the prompts mainly focus on the cultural attributes of images during the generation process, it becomes very challenging to embed these cultural features associated with the image content into the pre-trained model through fine-tuning. Notably, we can alleviate the potentially challenge by applying a fine-tuning workflow with task-specific dataset. Recent works have demonstrated the possibility of fine-tuning pre-trained models to other tasks, like vision tasks Dinh et al. (2022); Lu et al. (2023); Wu et al. (2023), NLP problems Bakker et al. (2022); Hu et al. (2023), and reinforcement learning area Reid et al. (2022). A common sense can be obtained from these approaches that fine-tuning format can address the issue between generality and specific-task in cross-domain learning. To be noticed, most diffusion fine-tuning mothods focus on image property while the annotations of these images played an equally important role since features of modal alignment are included in these annotations. In the text-to-image inference process, the conditional information mainly comes from an input text prompt, which can be a sentence consisting of objects or more abstract requirements Chen et al. (2024b).

In this paper, we focus on fine-tuning diffusion models combing LLM models for generating images with peculiar representation features in the Chinese ancient architecture area. Noted that the Chinese ancient buildings vary a lot for not only different appearance, but also different culture backgrounds. As shown in Fig. 1, the buildings of different dynasties carry some unique characteristics that Chinese architectural elements may share names with those in other cultures, such as roofs, beams, and courtyards. These features collectively contribute to the distinctive charm and enduring legacy of Chinese architectural heritage. On the other hand, information such as culture, geographical location, name, etc. cannot be intuitively presented in the content of the image, and these are important information of culture-related data. Compared with other similar buildings, the ancient ones exhibit uniqueness in terms of structural details, cultural significance, layout, materials, stylistic diversity and integration of natural elements. All these bring great challenge in the generative models as mentioned in the dreambooth Ruiz et al. (2023). As a result, the main challenges for generating models lie in accurately capturing in the training process and reproducing the differences including both content and culture information during the inference time. Addressing these challenges requires comprehensive multimodal datasets, fine-tuning diffusion and LLM models, and collaboration with cultural experts. There are some other ways to prevent language drift Lee et al. (2019); Lu et al. (2020) by renaming the subject with class-specific prior preservation loss as in Ruiz et al. (2023).

To obtain satisfied ancient Chinese buildings, our research and innovation focus on three stages of data, model, and designated generating scenarios, and finally successfully preserve and integrate of the uniqueness of the ancient Chinese architectures in a pretrained diffusion model. Based on prior dataset, we first build a multimodal interleaved dataset with curated & segmented images and high-quality annotations. For images, we leverage the notable successful SAM2 model Ravi et al. (2024) to obtain pure content images. In order to optimize the annotations, we redefine the image types and feature names of the ancient buildings dataset Biao.Li et al. (2024) combining with relevant cultural background information on the Internet celebrities of the Cultural Relics Bureau to ensure the reasonable features must be learned during training. To overcome the language drift issue, we leverage the semantic prior of SAM2 on the class that is embedded in the model and recheck with human feedback which encourages the model to generate diverse instances of the same class as our objective of preserving the uniqueness. Secondly, two fine-tuning strategies, full parameter fine-tuning and LoRA Hu et al. (2021), are introduced in our experiments to explore the performance of our research. As mentioned in the Hunyuan model Li et al. (2024), the coverage of the data categories in the training data crucial for training accuracy. Therefore, our models extract two fundamental categories, subject and style. The subject catogory learns from the processed ancient building images and explore the style part with other aesthetic images with carefully designed prompts. Finally, we adopt the outstanding Large Multimodal modal (LMM) BLIP3 Xue et al. (2024) and innovative LLM derived methods, RAG Fan et al. (2024) and GraphRAG Peng et al. (2024), to enable an accurate and comprehensive relational module capturing these unique attributes and underlying culture identity that set them apart. The RAG is leveraged for combing text from BLIP and collected domain culture information in the training stage. During inference stage, the GraphRAG model is used to enhance the quality of prompts with these domain background information. To evaluate our work, we compare the generated results and quantitative metrics with other outstanding models to proof the advantage of our model in the generating area of the Chinese ancient architectures. The generated results exhibits subject fidelity and prompt fidelity according to the data characteristics.

## 2 BACKGROUND

### 2.1 CONDITIONAL DIFFUSION MODEL

Most of the current image related diffusion models are conditional diffusion models, which are also the general basis for the implementation of cross-modal tasks. In details, a multimodal dataset consisting of sample pairs $(x^i, y^i)$, where the $x^i$ represents the image and $y^i$ expresses the corresponding label, are used to train a diffusion model. As mentioned in Chen et al. (2024b), the objective of the training is to estimate the conditional score function during the backward denoising process. The function is:

$$d\tilde{X}_t^{y,\leftarrow} = [\frac{1}{2}\tilde{X}_t^{y,\leftarrow} + \hat{s}(\tilde{X}_t^{y,\leftarrow}, y, T-t)]dt + d\bar{W}_t, \quad with \quad \tilde{X}_t^{y,\leftarrow} \sim N(0, I_D). \quad (1)$$

where the $\tilde{X}_t^{y,\leftarrow}$ is the training image with conditional annotation $y$ in the backward process $\leftarrow$. The $\hat{s}(x, y, t)$ is the estimator of the real score function $\nabla log p_t(x)$ which is the gradient of the log probability density function $X_t \sim P_t$. The $T$ refers to the total number of noise adding from clean sample to the pure noise and the $\bar{W}_t$ indicates a Wiener process. The function is used learn the correspondence between the image X and the annomation Y which can be further used to sampling from the conditional distribution $P(x = images|y = annomation)$.

In reality, conditional content can be various types of modalities, such as subject Radford et al. (2021), text prompt Podell et al. (2023), part of an image Kawar et al. (2022), depth image Zhang & Agrawala (2023), bioinformatics Guo et al. (2023), etc. Most of these conditional information $y$ is discrete and the score function $\nabla log p_t(x_t|y)$ can be parsed via the Bayes' rule into two parts,

$$\nabla log p_t(x_t|y) = \nabla log p_t(x_t) + \nabla log c_t(y|x_t). \quad (2)$$

The first part $\nabla log p_t(x_t)$ mainly focus on the image features and can be learning in the diffusion model by the unconditional score function. The other one $\nabla log c_t(y|x_t)$ is related to the conditional information, like image categories, and always leverage a pre-trained model, like CLIP, to capture the latent structure between $X$ and $Y$.

## 2.2 SAM AND BLIP

Both SAM Kirillov et al. (2023) and SAM2 Ravi et al. (2024) are highly successful image segmentation models with demonstrated performance in various scenarios. It is designed to generate a valid segmentation mask according to segmentation prompt including spatial or text information of subjects. In this paper, we choose the SAM2 model since there are a larger and diverser dataset containing images and videos which are used for training. There are mainly five components in the SAM2 model. The image encoder, which use an MAE He et al. (2022) pre-trained Hiera Ryali et al. (2023); Bolya et al. (2023) image encoder, provides feature embeddings for subsequent components. The memory attention is used to condition the current frame according to prior frames. The prompt encoder and mask decoder are used to define the extent of the object and predict multiple masks. The memory encoder downsample the output mask to provide memory to the last component - memory bank which retains information about past predictions for the target object in the video. With the SAM2, we can easily segment objects of interest in an image. The model exhibits strong generalization to unseen objects for the unseen task from a limited number of images. By filtering the mask, We can get a new pure content image with the background removed.

Beside the SAM2 model, how to generate the captions from images plays an important role in our Image-Annotation-Augmented Diffusion pipeline. The BLIP3 Xue et al. (2024) exhibits outstanding in-context learning capabilities compared with other open-source LMMs with similar model sizes. multimodal capabilities. The BILP3 facilitates the connection of pre-trained language models to visual inputs through lightweight connectors, streamlining the integration process while preserving strong multimodal functionality. To enhance the training of BLIP3, they leverage a diverse ensemble of multimodal and curated caption datasets, along with publicly available resources. Moreover, a scalable vision token sampler and simpler training objectives are introduced to refine the model architecture. The impressive results of BLIP3 demonstrates its emergent abilities such as multimodal in-context learning on many multimodal benchmarks. As a result, we choose the BLIP3 model as the model for generating text from images.

## 2.3 RAG AND GRAPHRAG

Recently, the Retrieval-Augmented Generation (RAG) Fan et al. (2024) has been widely used to address the hallucination Huang et al. (2023) issue that comes from the inaccurate or even fabricated information from LLMs. It comes from the missing corpus out of the pre-training dataset, such as domain-specific knowledge, real-time updated information, and proprietary contents. The RAG integrates a retrieval module to combine external knowledge with the language comprehension and text generation capabilities of LLMs. The RAG achieves impressive results and ensures factuality and credibility in various domain task performance with domain-specific information. In this paper, we leverage the RAG combining the official information from Shanxi Cultural Relics Bureau `https://wwj.shanxi.gov.cn/` to enhance the annotation of these ancient architectures. As mentioned in Peng et al. (2024), the RAG faces several limitations in real-world scenarios, including Neglecting Relationships, Redundant Information, and Lacking Global Information.

To enhance prompt words more efficiently and concisely, the GraphRAG Peng et al. (2024); Edge et al. (2024) emerges as the solution. A pre-constructed graph including knowledge of the Chinese ancient architecture is retrieved by the GraphRAG for a broader context and interconnections within these traditional architectural treasures and cultural connotations. The GraphRAG is a variant of RAG in graph data space of RAG and retrieves relevant relational knowledge, including nodes, triples, paths, and even subgraph, from a pre-constructed graph compared with the text corpus of RAG. As a result, GraphRAG is particularly suitable for tasks that have textual data that are related to each other. The relationships between texts and entities incorporate the structural information that is taken into account beyond the text message. Moreover, in the process of constructing graph-based data, raw textual data may be subjected to filtering and summarization procedures, thereby contributing to the enhanced refinement and accuracy of the information represented within the graph. The total process of learning the target distribution $p(a|q, \mathcal{G})$ can be formulated as:

$$p(a|q, \mathcal{G}) = \sum_{G \subseteq \mathcal{G}} p_\phi(a|q, G) p_\theta(G|q, \mathcal{G}), \tag{3}$$

where $a$ is the answer of the retriving query $q$ based on domain-specific graph $\mathcal{G}$. The $p_\phi(*)$ is the answer generator, like LLMs, and the $p_\theta(*)$ is the graph retriever.

# 3 METHOD

In this paper, our research focuses on the conditional information analysis of the Chinese ancient architecture. In this section, we first introduced the structural design of the entire Image-Annotation-Augmented Diffusion pipeline in 3.1. Then, we introduce the motivation and processing of images and their corresponding texts in 3.2 and 3.3, respectively.

## 3.1 OVERALL ARCHITECTURE

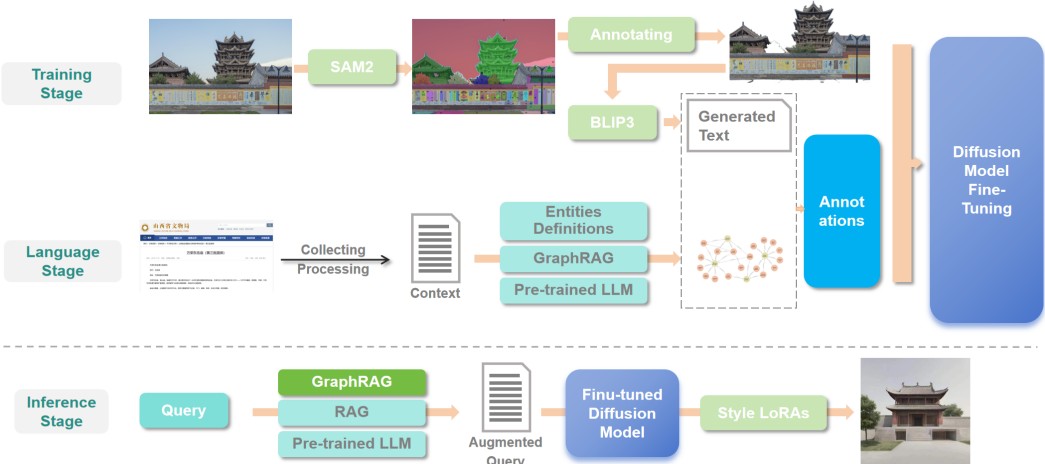

Figure 2: It is the overall architecture of our Image-Annotation-Augmented Diffusion pipeline. There are three major modules, including the training stage, the language stage and the inference stage. Since we fine-tune a pre-trained diffusion model, the language stage can be processed separately by further LLMs in both the training stage and the inference stage. Some leading models are used in our model, such as the SDXL, BLIP3, SAM2, GPT4o mini.

The overall architecture of our Image-Annotation-Augmented Diffusion pipeline is illustrated in the Fig. 2. Given a multi-modal dataset consisting of $N$ images and their corresponding annotations: a language description specifying the content, location, and culture backgrounds. The final goal of our Image-Annotation-Augmented Diffusion pipeline is to fine-tune a pre-trained Diffusion model for intergreting the unique representation $P(x_n = Image|y = Annotationlabel)$, including image features and text descriptions, of the Chinese ancient architecture.

Previous research Ruiz et al. (2023); Dong et al. (2023) did domenstrate that fine-tuning the pre-trained diffusion model, like SDXL, based on partial images can improve the generation ability with unique object characteristics in specific fields. Therefore, we chose to fine-tune a pre-trained diffusion model-SDXL Podell et al. (2023) for the Chinese ancient architecture dataset. In order to better verify the final embedded unique content, we leverage both global variable fine-tuning as in the Dreambooth Ruiz et al. (2023) and Low-Rank Adaptation (LoRA) Hu et al. (2021) fine-tuning strategy, respectively. Since the objective of our research is to implant the subject instance into the output domain of the diffusion model, the natural way is to fine-tune the model to integrate the visual features and semantic representations of the specific domain. To enhance parameter efficiency, the LoRA approach is introduced by freezing the pre-trained weight matrices of the pre-trained SDXL and integrating additional trainable low-rank matrices.

For the Image-Annotation-Augmented Diffusion pipeline, the fine-tuning of image and annotation should be trained simultaneously. In general, the whole procedure can be divided into three stages, as shown in the Fig. 2, including the Training Stage, Language Stage, and Inference Stage. It is worth noting that the text, as the key representation information for generating model fine-tuning, can be used multiple times in the model training and inference stages. Hence, we have specifically highlighted the language module as a separate stage. The goal of our research is to augment domain-

specific annotations into the text-to-image in a latent representation space, like the CLIP Radford et al. (2021), accompanying with the image features into a pre-trained diffusion model.

## 3.2 IMAGE PROCESSING

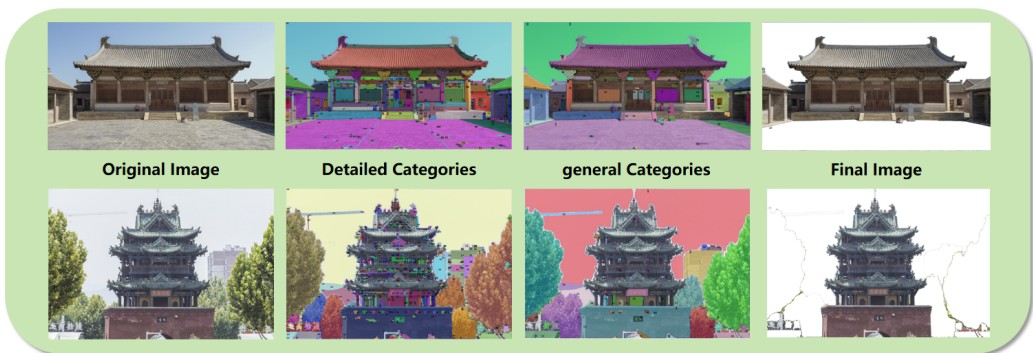

Figure 3: In this figure, we provide segmenting results of the SAM2 mdoel. With different hyperparameters, the final masklets exhibit different hierarchical results, including detailed categories and general categories. For the Chinese ancient architecture, our research foucs on the general entity which can be better aligned with semantics.

As mentioned in the Li et al. (2024), the data categories play the central role for training an accurate model. In general, there are two fundamental categories: Subject $x_{sub}$ and Style $x_{sty}$. Normally, these two categories are distinguished by the annotation information of the images. For example, the prompt *"Cartoon drawing of an outer space scene. Amidst floating planets and twinkling stars, a whimsical horse with exaggerated features rides an astronaut, who swims through space with a jetpack, looking a tad overwhelmed."* contains *"Cartoon drawing"* as the style description and the rest words as the subject description. Normally, the $x_{sub}$ simply describe the subject of the image $x$ and omit background details or the latent connections portrayed in the image.

In DALL-E 3 Betker et al. (2023), their research focuses on how to create a dataset of long, highly-descriptive captions. However, these text descriptions do not include the specific location of the content described in the image, or the accurate content, nor do they include information about the correlation between them. As a result, we changed our research ideas from enriching the description of the image content to condensing the information of the image, and trying to retain only the relevant entities of the description. An intuitive idea is to classify the pixels of the image by the semantic segmentation model. In this paper ,we choose the leading semantic segmentation model SAM2 Ravi et al. (2024) as our tool. Based on the content for fine-tuning in the training stage, we can hide the irrelevant pixel areas and find the corresponding pixel areas of different entities through the SAM model. Finally, the processed image only retains the entity Mask area corresponding to the description content.

As shown in Fig. 3, the Chinese ancient architecture images with the background are chosen to be the input of the SAM2 model. To be noticed, the final masklets vary greatly based on different settings. We carefully choose the hyperparameters to meet the requirements for clearly obtaining a building edge. To better exhibit the segmenting results, we compare the detailed and general results in Fig. 3. The detailed categories take the overall architectural style down to the level of each component. Although the results obtained are richer and detailed, for the task of text-to-image generation, it is too detailed to accurately align the semantic and image features in the latent space. Moreover, these unclear entity relationships are easier to introduce noise and thereby affect the final generation results. In contrast, the general categories involve all entities which are mostly divided into a whole state. As a result, it is more in line with the research embedding the specific content entities of an image.

### 3.3 ANNOTATION AND PROMPT ENHANCEMENT

The research of caption improvement is a hot topic in the text-to-image generation area. As mentioned in the DALL-E 3 Betker et al. (2023), the poor quality of the text and image pairing of the dataset results to the unsatisfied performance of the model. Most prior researches focus on how to enrich the description, and enrich the captions from the main subject $y_{sub}$ to its background, surroundings, the involving text in the image, styles, colorations, etc. However, there is no accurate correspondence between these descriptions and the corresponding pixel space in the image. Therefore, the outline of the entity cannot be accurately located, which will produce confusing results in training and inference.

To end the issue, we first perform semantic segmentation on the image content, retaining only the pixels of the main content, and obtain an image of pure content $x_{sub}$. To obtain the captions of our generation dataset, we first try to reversely obtain the descriptions $y_{sub}$ corresponding to the image content through the BLIP3 Xue et al. (2024) model. Since there is no similar Chinese ancient architecture data in the training dataset of the BLIP3, the output exhibits inaccurate results. Therefore, we re-check all generated captions through **_human feedback_** and revise the irrelevant content to be $y_{HF-sub}$.

## 4 EXPERIMENTS

Moreover, We collected the corresponding background information $y_{cul}$ of these ancient buildings from the Shanxi Cultural Relics Bureau website `https://wwj.shanxi.gov.cn/`. Because most of these buildings are China's national heritage, the cultural information is more important than the content of the building itself. These texts contain detailed information, including not only architectural information, but also geographical location, cultural background, national treasure status, etc. All of the information interprets the background of a architecture from a unique perspective, which will play a crucial role in future generation tasks. Finally, the combination of $y_{HF-sub}$ and $y_{cul}$ becomes the final annotations $y_{anno}$ of the Chinese ancient architecture, and it is consistent with the description of ordinary people's subjective cognition. Therefore, in this paper, we focus on how to leverage the LLMs and their derived tools to incorporate this background information into the conditions of the diffusion model, while avoiding the introduction of confusing misinformation.

In the training stage, we fine-tune a pre-trained diffusion models with the proposed dataset $(x_{sub}, y_{anno})$. The relationship between image features and text annotations can be learned by the model in a more powerful way and used in the downstream tasks. Moreover, these culture information $y_{cul}$ can be further used in the inference stage to enhance the prompts. We choose the GraphRAG method, as in Edge et al. (2024), for semantic Parsing (SP)-based Peng et al. (2024) generation. The proposed GraphRAG constructs a logical form (LF) graph corresponding to each query, which is then executed against the knowledge base to extract the correct related words for prompt enhancement.

### 4.1 IMPLEMENTATION DETAILS

**Dataset.** Since our research introduces the Image-Annotation-Augmented Diffusion pipeline which focuses on building a content-only Chinese ancient architecture with domain specific annotations. Based on a public dataset Biao.Li et al. (2024), which includes 581 high-quality images of the Chinese Ancient buildings, we design a content-based image augmented pipeline. In details, we first resize the short side of images to 1024 resolution. After extracting refined content images with SAM2 Ravi et al. (2024), we filter 449 images with clear segmentation for subsequent captioning. As shown in Fig. 4, the basic annotations of images are obtained by BLIP3 Xue et al. (2024). To enrich the culture content, we use the RAG strategy and human feedback operations for more accurate and richer annotations. Meanwhile, we use the GraphRAG to extract effective architecture entities from its cultural background as supplemental descriptions. Finally, we build a new dataset containing pure subject images, their backgrounding informations, and the refined annotations. The proposed dataset will be released after review.

**Experimental setting.** The experiments are implemented based on the pre-trained SDXL Podell et al. (2023). We utilize the proposed multimodal dataset for fine-tuning. Both full-parameters fine-tuning and LoRA strategy are adopted in our research. For full-parameters fine-tuning, the initial

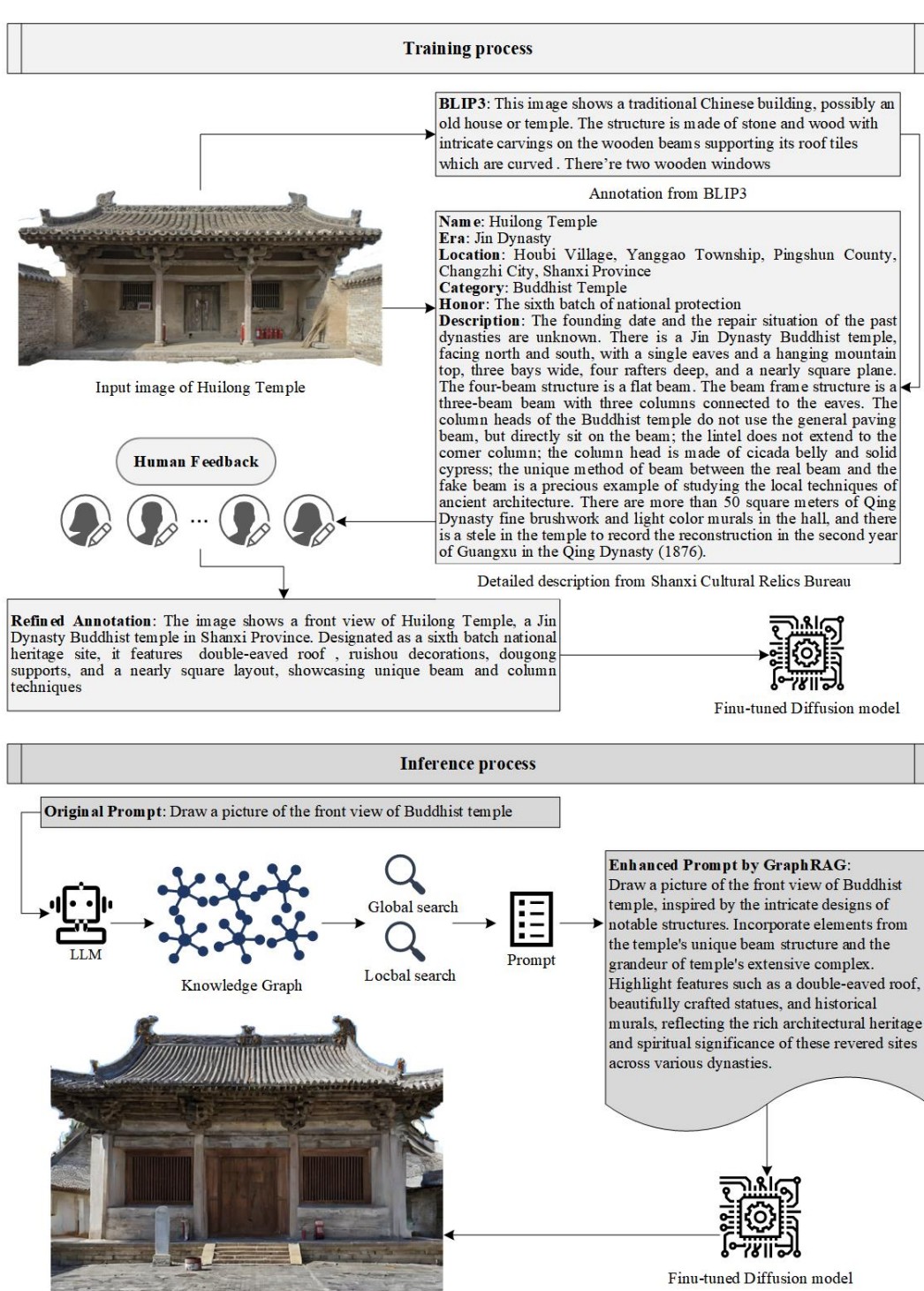

Figure 4: In this figure, we exhibit the annotation and prompt related content in the Image-Annotation-Augmented Diffusion pipeline. All mentioned models, such as BLIP3, RAG, Human Feedback processing and GraphRAG, are shown in both the training and inference stages.

learning rate is set as 1e-4 using cosine with restarts scheduler and the experiments are conducted on 2 A100 GPUs with fp16 precision and a total of 10000 steps. We applied Adam optimizer to

optimize parameters. For the LoRA strategy, we select 80 images and train a total of 10 epochs. More results are shown in the inference stage for evaluation.

## 4.2 COMPARISON AND ANALYSIS

As shown in the Fig. 4, the $X_{sub}$ images are used for fine-tuning. Based on their domain unique descriptions from BLIP3 $Y_{sub}$ and corresponding background information $Y_{cul}$, refined annotations are augmented in a LLM and human feedback way. During the training process, our multimodal dataset covers segmented images and captions refined by the RAG and human feedback. For inference stage, there is a better prompt combinng the original prompts provided by users and the GraphRAG refined prompts with specific backgrounds.

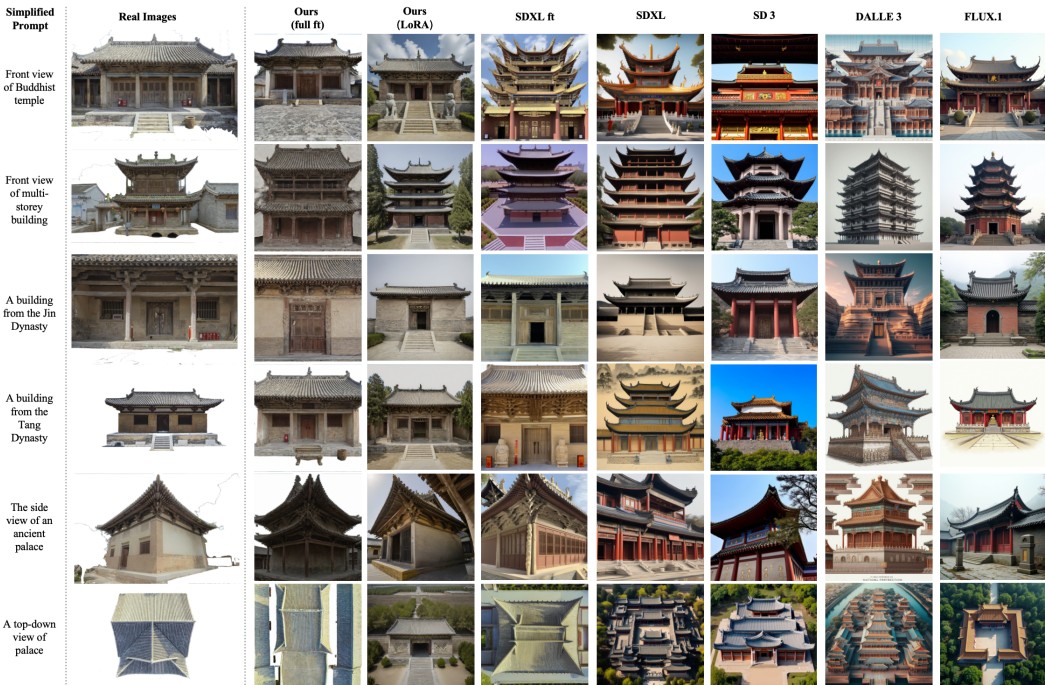

Figure 5: The comparison of our method and other state-of-the-art models, including SDXL, fine-tuning SDXL, SD3, DALL-E3 and FLUX.1, with our enhanced prompts. The prompts are enhanced by GraphRAG and cover different conditions like perspective view, architectural type and dynasty. For the limitation of space, we just provide some key words here and the complete prompts are shown in the appendix part.

To evaluate the performance of our proposed Image-Annotation-Augmented Diffusion pipeline, we randomly select six prompts from six angles, covering multiple views, building types and background information. Enhanced by the GraphRAG, the initial prompts are enhanced into prompts with rich connotation. We test two different fine-tuning methods, all parameters fine-tuned SDXL (full ft) and LoRA Hu et al. (2021), to evaluate our proposed new pipeline. As a result, we compare the generations with sevral models, including fine-tuned SDXL with the original dataset (SDXL-ft) Biao.Li et al. (2024), initial SDXL Podell et al. (2023), SD3 Esser et al. (2024), DALL-E 3 Betker et al. (2023) and FLUX.1 https://flux-ai.io/flux-ai-image-generator/ in Fig. 5. To be noticed, the enhanced prompts are reduced to simple key words for the limitation of space. We provide all six complete prompts in the appendix part.

In order to verify the results intuitively, we display some reference images from the proposed dataset in the first column. It can be observed that all these results capture the basic form of Chinese ancient architectures. However, the generation results without fine-tuning (SDXL, SD3, DALLE3, FLUX.1) exhibit two obvious drawbacks. The first issue is the lack of structural variety, as most imitate the ancient buildings of the Forbidden City. The second problem is that the images do not match the text very well. In contrast, the fine-tuned SDXL (SDXL ft) maintain the characteristic of ancient

architectures, including architectural style, color and texture. However, SDXL-ft suffers from the alignment between refined prompts and results. For example, when we attempt to generative a Chinese Buddhist temple, the result display a temple similar to those found in Thailand, even though we specified Chinese architecture. Meanwhile, our method shows better results in content quality and image-text alignment.

We further quantitatively evaluate the performance of the proposed model, which aim to explore the specific-area generation task using Image-Annotation-Augmented Diffusion pipeline. However, many general evaluation models, like LAR-IQA Avanaki et al. (2024) and ImageReward Xu et al. (2024), are not suitable for the domain-specific application. One reason comes from the point that the evaluation criteria for these methods are trained on large-scale general datasets as a blackbox. However, our research focuses on the ancient architectural content generation rather than the overall style. Therefore, we conduct the comparing experiments with the FID Seitzer (2020) and the Clean-FID Parmar et al. (2022), which compute the distribution difference between generated images and real ancient architectures. Specifically, we use the image dataset segmented by SAM2 (FID 1 & Clean-FID 1) and original dataset (FID 2 & Clean-FID 2) to calculate the FID similarity with the generated results, which can more accurately evaluate the degree of content preservation of ancient buildings.

| | SD3 | DALLE 3 | FLUX.1 | SDXL | SDXL ft | Ours(full ft) | Ours(LoRA) |
|---|---|---|---|---|---|---|---|
| FID 1 ↓ | 196.34 | 236.72 | 193.30 | 225.81 | 173.30 | 174.33 | **153.26** |
| FID 2 ↓ | 194.39 | 236.28 | 189.28 | 220.23 | 167.81 | 175.75 | **147.28** |
| Clean-FID 1 ↓ | 200.01 | 252.76 | 196.11 | 214.77 | 175.46 | 174.43 | **152.34** |
| Clean-FID 2 ↓ | 197.68 | 252.30 | 191.53 | 209.98 | 171.81 | 176.55 | **146.01** |

Table 1: Quantitative evaluation on the difference between real ancient Chinese buildings and generative results from state-of-the-art methods.

In Table 1, both of our methods, full ft and LoRA, achieve satisfied scores. The LoRA gets the best performance among all models. It is also evidence that a large number of fine-tuning uses the LoRA method in reality. The results of SDXL-ft are close to ours in these metrics, which is due to the consistency of the dataset. However, it can still be found that our model outperforms SDXL-ft in image-text alignment as shown in Fig. 5. One fact shows that the recent famous FLUX.1 achieves the best performance among four un-finetuned models which is consistent with user's experience.

In the experiment, we mainly discuss preserving the content of ancient architecture in pre-trained models with the SAM for image features and BLIP3, RAG and GraphRAG for annotations, without focusing on the overall texture and background details. The comparative experiments demonstrate that our method can effectively retain content in SDXL model. In the future, we will continue to explore that our method can separate content and style for seamless integration. Moreover, we will fine-tune the FLUX.1 model to verify the effectiveness of our method.

## 5 CONCLUSION

In this paper, our research focuses on the generation task of the Chinese ancient architecture. To preserve the unique heritage, both images and their annotations are enhanced with different treatments, including retaining subject area of the image through semantic segmentation and using RAG and GraphRAG strategies to embed cultural information and form correlations in the latent space. By combining content and style differentiation, and incorporating models like SAM2, BLIP3, RAG and GraphRAG, we ensure the generated images are both culturally accurate and visually precise. This work highlights the potential of fine-tuning AI models for specialized tasks, paving the way for further developments in culturally-aware image generation.

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
