The exhibition of prompts is list as follows. We provide the simple input prompt, the GraphRAG enhanced prompts and the simplified prompts in the Fig.5.

| | Initial Prompt | GraphRAG Enhanced Prompt | Simplified Prompt |
|---|---|---|---|
| 1 | Draw a picture of a building related to Buddhist Temple | the front view of Buddhist temple, inspired by the intricate designs of notable structures. Incorporate elements from the temple's unique beam structure and the grandeur of temple's extensive complex. Highlight features such as a double-eaved roof, beautifully crafted statues, and historical murals, reflecting the rich architectural heritage and spiritual significance of these revered sites across various dynasties. | Front view of Buddhist temple |
| 2 | Draw a picture of a building from Song Dynasty | the front view of multi-storey Pagoda building from the Song Dynasty. This octagonal, seven-story structure, showcasing dense eaves and a unique shrinking design. Its hollow first to third floors reflect the architectural techniques of the era. As part of the Yuanqi Temple complex, this pagoda not only serves as a Buddhist temple but also embodies the rich cultural heritage and historical significance preserved through various renovations across dynasties. | Front view of multi-storey building |
| 3 | Draw a picture of a building from Jin Dynasty | a building from the Jin Dynasty, such as the significant Buddhist site located in Shanxi Province. There is a door in the middle and the building is symmetrical. This temple, recognized as part of China's national heritage, showcases distinctive architecture featuring a single-eaved hanging mountain top design. Temple reflects the era's artistic styles and religious practices, making it a remarkable representation of Jin Dynasty architecture and cultural heritage. | A building from the Jin Dynasty |
| 4 | Draw a picture of a building from Tang Dynasty | Draw a building from the Tang Dynasty, a significant Buddhist temple located in Shanxi Province. This architectural marvel, recognized as one of China's first national protection sites, showcases a distinctive single-eaved hip roof covered with barrel tiles. Built on a stone platform, it features a large Buddha altar adorned with colorful Tang Dynasty sculptures and intricate exposed beams, preserving the era's unique construction techniques and cultural heritage. | A building from the Tang Dynasty |
| 5 | Draw a picture of a building honred as National Protection | The side view of an ancient Palace. This palace showcases traditional Chinese architecture with intricate roof designs and historical steles, reflecting its cultural significance. Each building embodies rich history and artistry, deserving of recognition and preservation. It was honored as National Protection. | The side view of an ancient palace |
| 6 | Draw a picture of a building like Yongle Palace | a top-down view of palace, a significant cultural site in Shanxi Province, China. Known for its Folk Taoism roots, this architectural marvel from the Yuan Dynasty. The palace is celebrated for its stunning Yuan Dynasty murals, showcasing exceptional artistry. Recognized as a national protection site, It embodies the rich cultural heritage and architectural brilliance of its era. | A top-down view of palace |