# OpenReview forum: "Preserving the Unique Heritage of Chinese Ancient Architecture in Diffusion Models with Text and Image Integration"
_ICLR.cc/2025/Conference — Submitted to ICLR 2025_

### Official Review · Reviewer_Uzyh · 2024-10-23

**Soundness:** 2
**Presentation:** 3
**Contribution:** 2
**Rating:** 5
**Confidence:** 4

**Summary:**

In this paper, the authors present an Image-Annotation-Augmented Diffusion pipeline integrated with human feedback. This pipeline aims to explore the specific-area paradigm for image generation considering small data amounts and professional concepts. Initially, the Segment Anything 2 (SAM2) model is utilized to acquire refined content images, enabling an in-depth examination of the relationship between unique characteristics and multimodal image generation models. Subsequently, the Retrieval-Augmented Generation (RAG) and BLIP3 are employed to obtain high-quality annotations. Through these steps, the fine-tuned model demonstrates the enhanced performance of the proposed pipeline compared to existing models. Experiments confirm the pipeline's effectiveness in preserving and integrating the unique features of ancient Chinese architecture.

**Strengths:**

1. The writing is clear and understandable, facilitating comprehension of the research content and methods.

2. The experimental results appear promising, indicating the potential of the proposed pipeline in handling the specific task of generating images related to ancient Chinese architecture.

**Weaknesses:**

The research domain is relatively narrow, focusing specifically on ancient Chinese architecture. This limited scope might pose a challenge for broader acceptance at a conference like ICLR, despite the seemingly good performance within the specific domain.

**Questions:**

1. Why not attempt to apply the proposed pipeline to other domains to further verify its effectiveness?
2. Are there any potential limitations or challenges that might arise when applying this pipeline to other types of cultural or architectural heritages?
3. How might the performance of the pipeline be affected by different levels of data availability and quality in other domains?

---

### Official Review · Reviewer_TkwJ · 2024-11-01

**Soundness:** 3
**Presentation:** 3
**Contribution:** 3
**Rating:** 6
**Confidence:** 3

**Summary:**

This paper introduces an Image-Annotation-Augmented Diffusion pipeline that preserves the unique heritage of Chinese ancient architecture in diffusion models. The process begins by using SAM2 to accurately crop architectural objects from images, followed by generating text descriptions using the image captioning model BLIP3. Next, RAG, GraphRAG, and LLM models process professional texts sourced from the Cultural Relics Bureau to identify relationships between architectural elements. Finally, human feedback is applied to refine the text, producing the final text prompt for the diffusion model.

**Strengths:**

1. This paper focuses on fine-tuning a diffusion model for synthesizing images of Chinese ancient architecture, which is beneficial for the digital preservation of historical relics and the advancement of related industries.

2. The pipeline is well-structured and practical. It considers both the subject and style of the architecture during model training. Additionally, combining RAG methods and LLM with human feedback improves the accuracy of the text descriptions.

**Weaknesses:**

1. This paper resembles an engineering project that applies existing methods and pipelines for generating images of Chinese ancient architecture. Regarding methodological novelty, the combination of these existing methods is not particularly innovative. However, the proposed pipeline effectively addresses the unique features of Chinese ancient architecture and successfully achieves accurate image generation.

2. The experimental testing could be strengthened. For instance, attributes like dynasty, cultural style, and location are valuable for characterizing architectural features. During inference, the authors could vary one or more of these attributes to observe differences in the generated architecture. Such testing would help verify whether the learned diffusion model accurately understands key architectural information.

**Questions:**

1. In Figure 5, the generated image of Ours (LoRA) seems to depict an incorrect view for the prompt“A top-down view of palace.”

2. In Figure 1, unique feathers --> unique features?

---

### Official Review · Reviewer_3fWp · 2024-11-02

**Soundness:** 2
**Presentation:** 1
**Contribution:** 2
**Rating:** 3
**Confidence:** 5

**Summary:**

This paper aims to use diffusion models to generate Chinese ancient architecture with specific features or styles. The author uses LoRA to fine-tune a pretrained diffusion model, and proposes "Image-Annotation-Augmented Diffusion" pipeline that combines image segmentation, annotations enhanced by human feedback, and domain-specific cultural data. The approach involves using SAM2 for precise image segmentation, BLIP3 for text annotations, and GraphRAG modules to retrieve and augment cultural data. The experimental results show effectiveness of fine-tuning methods on diffusion models.

**Strengths:**

1)	The topic is interesting and meaningful. The author tries to fine-tune diffusion models to work on specialized domain, such as Chinese ancient architecture.
2)	The author introduces several models, such as SAM2 and BLIP3, to help improve the quality of image-caption pairs, so as to fine-tune the diffusion model to generate accurate outputs.
3)	The experimental results show effectiveness of the method that can reduce the FID of Chinese ancient architecture images generated by diffusion models.

**Weaknesses:**

1)	The representation is poor. There are many spelling mistakes and grammatical mistakes in the paper, such as “mothods”, “catogory”, “... to proof the advantage” and “The generated results exhibits ...”. Therefore, I question the completeness of the paper.
2)	The effectiveness of some experimental results is not obvious. For example, Fig. 5 shows the comparison of the method with other models. However, there is little difference between the outputs of fine-tuned SDXL and the paper, and the output image of paper’s method with LoRA in line 5 is not reasonable. Also, the style of reference pictures is too monotonous, which is not enough to prove the effectiveness of the method in this paper to achieve the goal of generating buildings with various specific styles.
3)	The lack of novelty. I appreciate that the author uses SAM and other models for processing images and annotation to obtain better data quality. However, there is not enough contribution to innovation methods.

**Questions:**

What if I want to generate Chinese ancient architecture with other specific styles accurately?

---

### Official Review · Reviewer_UZaR · 2024-11-03

**Soundness:** 2
**Presentation:** 2
**Contribution:** 2
**Rating:** 3
**Confidence:** 4

**Summary:**

The paper introduces a pipeline for generating images of Chinese ancient architecture by preserving cultural and structural details in generated images. By combining many existing techniques, the proposed method can capture architectural elements in fine detail. The authors compare the proposed method with pre-trained text-to-image models, showing the effectiveness of the proposed method in image-text alignment and preservation of unique architectural traits.

**Strengths:**

- The research has potentials in architectural heritage preservation.
- Use of RAG and GraphRAG for enhancing text prompts with cultural background improves alignment between image content and historical context.

**Weaknesses:**

- The novelty is limited, which is not enough for ICLR.
- Contributions are unclear. The proposed method is just a combination of existing techniques, which is not sufficient for ICLR.
- The paper writing is not good and need to be revised.
- The comparison is unfair. The proposed method is finetuned while compared methods are based on pre-trained models.

**Questions:**

- The paper should be revised to correct English writing mistakes and typos.
- What is the complexity of the proposed method when the authors combine many techniques?

---

### Meta-Review · Area_Chair_BQZX · 2024-12-19

**Metareview:**

In this work, most reviewers vote for rejection, considering the limited applications for ancient Chinese architecture, the limited novelty of combining several existing modules, and bad writing. Thus, it is hard for AC to accept this paper.

### Summary
The paper introduces a diffusion pipeline for preserving ancient Chinese architecture's cultural and structural uniqueness.

### Pros
1. **Domain-Specific Contribution**: Focuses on the underexplored area of ancient Chinese architecture.
2. **Methodical Framework**: Combines existing tools effectively for enhanced data quality and alignment.
3. **Human Feedback Integration**: Improves annotation accuracy and cultural relevance.

### Cons
1. **Limited Novelty**: Combines existing techniques with minimal innovation.
2. **Narrow Focus**: Restricts applicability to Chinese architecture only.
3. **Poor Presentation**: Contains numerous grammatical errors and unclear contributions.
4. **Experimental Gaps**: Results lack diversity and fail to validate broader applicability.
5. **Unfair Comparisons**: Evaluate fine-tuned models against baselines without similar adaptation.

**Suggestions**: Broaden the scope to other architectural styles, improve writing quality, clarify contributions, and strengthen experiments with diverse and unbiased comparisons.

**Additional Comments On Reviewer Discussion:**

In the discussion, reviewers share the limited novelty of the specific applications. Even though the authors try to highlight their contributions,  most reviewers disagree, as does the AC. The details are as follows:

#### Points Raised by Reviewers:
1. **Limited Novelty** (UZaR, 3fWp, TkwJ): The reviewers emphasized the lack of innovation, noting that the pipeline primarily combines existing methods without significant methodological advancements.
2. **Narrow Scope** (TkwJ, Uzyh): The paper's focus on Chinese ancient architecture limits its broader applicability and relevance to the ICLR audience.
3. **Writing and Presentation Issues** (UZaR, 3fWp): Reviewers highlighted numerous grammatical errors and poor representation, which negatively impacted readability and professionalism.
4. **Experimental Weaknesses** (3fWp, TkwJ): Concerns were raised regarding insufficient diversity in experimental results and the limited evaluation of different styles or attributes.
5. **Unfair Comparisons** (UZaR): The proposed fine-tuned model was compared to baselines without similar adaptations, potentially skewing results.

#### Author Responses:
1. **Novelty**: The authors clarified that their contribution lies in integrating existing methods to address domain-specific challenges, particularly in data-scarce areas. They also highlighted their dataset creation and annotation improvements as contributions.
2. **Scope**: The authors acknowledged the narrow focus and proposed future work to apply the pipeline to other domains, such as Gothic architecture or traditional paintings.
3. **Writing and Presentation**: The authors committed to revising the manuscript to address grammatical errors and improve clarity.
4. **Experimental Weaknesses**: They explained limitations in style diversity due to dataset constraints and promised more comprehensive testing in future work.
5. **Comparisons**: The authors acknowledged the unfair comparisons and clarified that additional baselines would be included in future revisions.

The responses partially addressed the reviewers’ concerns, particularly regarding writing and presentation. However, the core issues of limited novelty and narrow applicability remained insufficiently resolved. While the authors’ clarifications and commitments for future improvements are appreciated, the lack of significant innovation and broader relevance outweighed these efforts. Thus, the decision is rejection.

To satisfy the quality of top conference papers, the author should highlight the unique difficulties of generating ancient architectures and propose modules for this problem. Furthermore, international ancient architectures, instead of only Chinese ones, are more convincing in showing the merit of the proposed method.

---

### Decision · Program_Chairs · 2025-01-22

Reject